# Water Spinach (*Ipomoea aquatica* F.) Effectively Absorbs and Accumulates Microplastics at the Micron Level—A Study of the Co-Exposure to Microplastics with Varying Particle Sizes

Yachuan Zhao [1,2], Can Hu [1,2,*], Xufeng Wang [1,2], Hui Cheng [1,2], Jianfei Xing [1,2], Yueshan Li [3], Long Wang [1,2], Tida Ge [4], Ao Du [1,2] and Zaibin Wang [1,2]

1   College of Mechanical and Electrical Engineering, Tarim University, Alar 843300, China; 10757223131@stumail.taru.edu.cn (Y.Z.); wxf@taru.edu.cn (X.W.)
2   Key Laboratory of Modern Agricultural Engineering of Colleges and Universities, Department of Education of Xinjiang Uygur Autonomous Region, Alar 843300, China
3   Department of Educational Administration, University of Saskatchewan, Saskatoon, SK S7N0X1, Canada
4   State Key Laboratory for Managing Biotic and Chemical Threats to the Quality and Safety of Agro-Products, Institute of Plant Virology, Ningbo University, Ningbo 315211, China; getida@nbu.edu.cn
*   Correspondence: 120140004@taru.edu.cn

**Abstract:** The absorption of microplastics (MPs; size < 5 mm) by plants has garnered increasing global attention owing to its potential implications for food safety. However, the extent to which leafy vegetables can absorb large amounts of MPs, particularly those > 1 μm, remains insufficiently demonstrated. To address this gap in knowledge, we conducted water culture experiments using water spinach (*Ipomoea aquatica* F.) as a model plant. The roots of water spinach were exposed to a mixed solution that contained fluorescently labeled polystyrene (PS) beads with particle sizes of 200 nm and 1 μm for 10 d. We utilized laser confocal scanning microscopy and scanning electron microscopy to record the absorption, migration, and patterns of accumulation of these large particle sizes of MPs within water spinach. Our findings revealed that micron-sized PS beads were absorbed by the roots in the presence of submicron PS beads and subsequently transported through the exosomes to accumulate to significant levels in the leaves. Short-term hydroponic experiments further indicated that high concentrations of PS bead solutions significantly inhibited the growth of water spinach owing to their large specific surface area that hindered the uptake of water and nutrients by the roots. In conclusion, both sizes of PS beads were found to be absorbed by water spinach, thereby increasing the risk associated with direct human consumption of microplastics in fruits and vegetables. This study provides valuable scientific insights to assess the pollution risks related to fruits and vegetables, as well as ensuring vegetable safety.

**Keywords:** microplastics; MP pollution; absorption; accumulation; water spinach; co-exposure experiment





## 1. Introduction

With the growing concern over MP pollution, there is increasing recognition of the processes by which soil MPs are absorbed and accumulated by plants [1]. MP particles that originated from agricultural activities have become prevalent in farmland soil, and potentially impact the ecological environment and compromise soil safety [2–4]. In comparison to the soil environment, MPs can enter plant tissues and be transferred to their edible parts, thereby potentially transferring accumulated MPs through the food chain to higher trophic levels [5–7]. For human health, directly edible hydroponic and soil-grown vegetables can absorb microplastics, particularly those that can migrate, transform and accumulate in the vegetables. Once ingested directly by humans or animals, MPs can have a range of toxic effects on organisms [8], such as the digestive system [9], respiratory system [10], nervous system [11] and reproductive system [12].

In the study of plant absorption of MPs, researchers have initiated relevant experiments and obtained partial confirmation. Li et al. [13] initially demonstrated the ability of plants to absorb MPs. However, there is still no definitive and consistent conclusion regarding particle size and the mechanisms of absorption. Through exposure to PS beads with a single particle size, cucumber (*Cucumis sativus* L.) plants were found to absorb MPs by their roots, with the maximum capacity of adsorption observed at MPs of 700 nm [14]. Current studies primarily focus on the absorption of nano/submicron-scale MPs by plants since micron-scale MPs are generally considered to be too large to penetrate the natural physical barrier of plants. This is particularly true for small vegetable plants. Li et al. [15] discovered that lettuce (*Lactuca sativa* L.) roots can take up MPs < 200 nm in size. Alternatively, Gao et al. [16] exposed the roots of water spinach (*Ipomoea aquatica* F.) to 80 nm PS beads, which resulted in their adsorption onto root epidermal cells rather than internalization or upward migration and accumulation. Conversely, the findings of Yu et al. [17] suggest the potential absorption of 500 nm polyethylene microplastics (PE-MPs) by cabbage (*Brassica oleracea* L. *var. capitata*) roots but lack effective confirmation.

Based on these findings, it has been acknowledged that nano/submicron MPs can be absorbed and accumulated by certain plant species. However, further study is still warranted to substantiate the potential absorption of MPs by edible leafy vegetables. In particular, there is a dearth of pertinent experimental evidence regarding the widespread absorption and accumulation of MPs with larger particle sizes ($\geq 1$ μm) in vegetables.

Herein, we selected water spinach as a representative leafy vegetable to investigate the universality of MP absorption in such crops, and explore the potential for absorbing larger particle sizes of MPs. Our hypothesis is that water spinach, which is an aquatic vegetable with a high capacity for absorption, would take up submicron-scale MPs. In addition, we hypothesized that once the pathway for entry of MPs into roots is established, it could also absorb micron-scale MPs. To test this hypothesis, we exposed the widely cultivated tropical and subtropical aquatic vegetable water spinach to a mixture of 200 nm and 1 μm PS beads for 10 d. We recorded the absorption and accumulation of these microplastics in its roots, stems, and leaves to achieve the following objectives: (1) confirm the universality of MP absorption in water spinach; (2) demonstrate the rapid migration and accumulation of absorbed MPs within the crop; and (3) establish that water spinach can take up MP particles $\geq 1$ μm. The findings from this study present novel requirements for the safety assessment of vegetables intended for human consumption.

## 2. Materials and Methods

### 2.1. PS Beads

Fluorescence-labeled and unfunctionalized PS-MPs with sizes of 200 nm and 1 μm were provided by Haian Zhichuan Battery Materials Technology Co., Ltd. (Nantong, Jiangsu, China). The 200 nm PS-MPs were labeled and dyed with fluorescein isothiocyanate (FITC), with excitation/emission wavelengths of 488/525 nm. The 1 μm PS-MPs were labeled and dyed with rhodamine 6G, with excitation/emission wavelengths of 525/580 nm. To avoid dye leakage, the PS microspheres were subjected to hydrophobic modifications. A laser particle size analyzer (Anton Paar GmbH, Graz, Austria) was used to determine the particle size distribution of the two differently sized microspheres. Based on the results, the standard particle size of FITC-labeled microspheres was $200 \pm 150$ nm, with approximately 23% of the PS microspheres at a size of 200 nm. The standard particle size of rhodamine 6G-labeled microspheres was $1 \pm 0.2$ μm, with approximately 27% of 1 μm PS microspheres (Figure S1a,b). The PS microspheres were prepared by a reverse phase emulsion polymerization method, which involved the addition of ultrapure water and trace amounts of SDS as dispersant to all the PS microspheres at a concentration of 50 mg L$^{-1}$. For the preparation of PS microsphere solutions, the 50 mg L$^{-1}$ PS microsphere solution was treated ultrasonically for 30 min and then diluted to 30 mg L$^{-1}$, 20 mg L$^{-1}$, and 10 mg L$^{-1}$ target concentrations. The microsphere morphology was characterized by

scanning electron microscopy (SEM) (Figure S1d,e), and the composition was determined by Fourier transform infrared spectroscopy (FTIR) (Figure S1c).

## 2.2. Plant Materials and Cultivation Conditions

The vegetable used in the experiments was water spinach, which was provided by the Lanpo Bay Vegetable Greenhouse (Alar, Xinjiang, China). The seeds were germinated in the greenhouse, transferred to pots after 10 d, and brought back to the laboratory to adapt to the growth conditions. Humus was used as the growth medium (Figure S2). After 4 d of growth in the laboratory, the roots and stems were washed with deionized water until no soil was visible, and the cleaned roots and stems were then transferred to hydroponic culture. The hydroponics medium was a mixture of ultrapure water and Hoagland's nutrient solution. Solutions of PS beads, of two bead sizes, with concentrations of 0 mg $L^{-1}$, 10 mg $L^{-1}$, 20 mg $L^{-1}$, 30 mg $L^{-1}$, and 50 mg $L^{-1}$, were added to the hydroponics. Ultrapure water was added every 2 d to keep the roots fully submerged. The plant growth conditions are shown in Figure S8. The plants were grown for 10 d under a 12/12 h light/dark cycle at $25 \pm 2$ °C. The experiment was repeated three times.

## 2.3. Transmission Electron Microscopy

Transmission electron microscopy (TEM) (JEM-2100Plus, JEOL, Tokyo, Japan) was used to observe the microfractures between cell walls in the water spinach cells grown under natural conditions. After the water spinach was cultivated for 10 d in a PS microbead solution at a concentration of 0 mg $L^{-1}$, the plants were washed with ultrapure water. The primary root was then cut off using a razor blade. The excess root hairs on the primary root were removed; the stem was cut off, and a section that was approximately 1 cm long was cut from the stem at a distance of approximately 10 cm from the junction of root and stem. The leaves were excised at the junction of the mesophyll and main vein. The small root, stem, and leaf tissue pieces cut with a razor blade were quickly placed in culture plates at 4 °C that contained 2.5% glutaraldehyde. They were then cut into tissue pieces $\leq 5$ mm using a razor blade and placed into centrifuge tubes that contained 2.5% glutaraldehyde. They were then incubated at 2–8 °C for 24 h. Before their placement in centrifuge tubes, a simple vacuum treatment was performed using a syringe to ensure the complete immersion of the plant tissue in the solution. After 24 h, the plant tissue samples were transferred to a fume hood and washed three times with phosphate-buffered saline (PBS) for 15 min each time. The samples were then placed in a solution of osmium tetroxide and PBS (1:1) in which they were immersed for 2 h. They were then washed three times with PBS for 10 min each. Subsequently, an alcohol gradient dehydration was conducted twice in 30%, 50%, 70%, 80%, 90%, and 100% ethanol for 10 min each. After dehydration, the plant tissues were embedded. The tissues were transferred to a 1:1 solution of acetone and ethanol for 10 min and then treated with pure acetone twice for 8 min each time. The tissue samples were then treated with a 1:3 solution of pure Epon 812 and acetone for 30 min, followed by a 1:1 solution of pure Epon 812 and acetone for 1 h, and left overnight in pure Epon 812. After embedding, gradient temperature polymerization was performed at 35 °C, 45 °C, and 60 °C for 3 h, 4 h, and 48 h, respectively. Finally, ultrathin sections were cut to a thickness of 50 nm using an ultrathin slicer (Leica EM UC7; Leica, Wetzlar, Germany). The samples were observed using a TEM at an acceleration voltage of 120 kV.

## 2.4. SEM

After water spinach had been cultivated in a solution of 50 mg $L^{-1}$ PS microspheres for 10 d, two-week-old plants were thoroughly cleaned with PBS. A razor blade was then used to cut off the primary, secondary, and tertiary roots. At the junction point with the roots, a 1 cm long stem segment was excised approximately 10 cm away from the root-stem junction. The leaf tissues were cut at the junction of the leaf blade and the main vein of the leaf. Small pieces of root, secondary root, tertiary root, stem, and leaf tissues were cut with a razor blade and quickly placed in Petri dishes at 4 °C that contained 2.5% glutaraldehyde.

Subsequently, the tissue pieces were further cut into pieces $\leq 5$ mm $\times$ 5 mm using a razor blade, placed in a centrifuge tube that contained 2.5% glutaraldehyde, and stored at 2–8 °C overnight. Before they were placed in centrifuge tubes, a simple vacuum treatment was performed using a syringe to ensure that the plant tissue was completely immersed in the solution. Alcohol gradient dehydration was then performed, which was similar to the treatment for the TEM samples. At least three plants per treatment group were examined. After drying, the samples were sprayed with an ion sputtering instrument (Cressington 108; Cressington Scientific Instruments UK, Watford, UK) for 60 s (with gold particles of approximately 1 nm in diameter) and wrapped with conductive tape to fix them on the sample stage and increase their conductivity. An SEM (SU3500, Hitachi, Tokyo, Japan) was used to examine the samples at an acceleration voltage of 10 kV. A field emission SEM (Apreo S; Thermo Fisher Scientific, Waltham, MA, USA), operating in the vacuum mode, was used to observe the samples at an acceleration voltage of 5 kV.

### 2.5. Laser Scanning Confocal Microscopy

Water spinach plants treated with 0 mg $L^{-1}$ and 50 mg $L^{-1}$ PS beads were selected. After their primary and lateral roots had been cut off, a section of the stem was excised at approximately 10 cm from the root junction. The nearest leaf to the root junction was selected. The samples were washed thoroughly with distilled water and quickly frozen in liquid nitrogen. They were then embedded in optimal cutting temperature compound (OCT) and cut into frozen sections of 7 µm, 20 µm, and 50 µm using a constant temperature freezing microtome (SYD-K2040, Yude Technology (Xiamen) Co., Ltd., Xiamen, China). The sections were observed using a laser scanning confocal electron microscope (A1R HD25; Nikon, Tokyo, Japan) with an increased filter and adjusted laser intensity and signal intensity until the PS bead fluorescence signal could be distinguished from the autofluorescence signal of the plant. At least three plants per treatment group were examined. Image capture and processing were performed using the NIS-Elements software (https//www.microscope.healthcare.nikon.com/zh_CN/products/software (accessed on 18 June 2023)).

### 3. Results and Discussion

#### 3.1. Larger Size PS-MPs Are Absorbed under the Influence of Submicron PS-MPs

The word "control" in all figures represents 0 mg $L^{-1}$ of MPs treated water spinach. No fluorescence signals were detected in the roots, stems, and leaves of water spinach treated with 0 mg $L^{-1}$ PS beads (Figures 1a and 2a). Moreover, SEM images showed that there were no spherical particles of 200 nm and 1 µm in the roots (Figure 3c,e), stems (Figure 4c,e), and leaves (Figure 5c). However, strong green fluorescence signals were observed on the root surface in the laser confocal images (Figure S3b,c), which indicated that a large amount of 200 nm PS beads were adsorbed on the surface of secondary roots. Moreover, the red fluorescent signal was primarily detected in the root tip region (Figure S3e,f). We hypothesize that the active cell division in the root tip elongation zone may have facilitated the penetration of root epidermal cells by PS-MPs. In the main root transverse section, approximately 20 cm away from the root tip, strong green, fluorescent signals were observed on the root surface (Figure 1a). Similar fluorescent signals were also observed in the central stele, while the signals were not as strong in the cortex. We also observed similar red fluorescent signals (Figure 1a), but their signal intensity was much weaker compared to those of the green fluorescence. The submicron-scale PS-MPs were more mobile than the micron-scale ones. Both fluorescence signals were primarily spread along the surface of roots and between root epidermal cells. In addition, we observed the transverse section of the main root at a distance of approximately 3 cm from the stem.

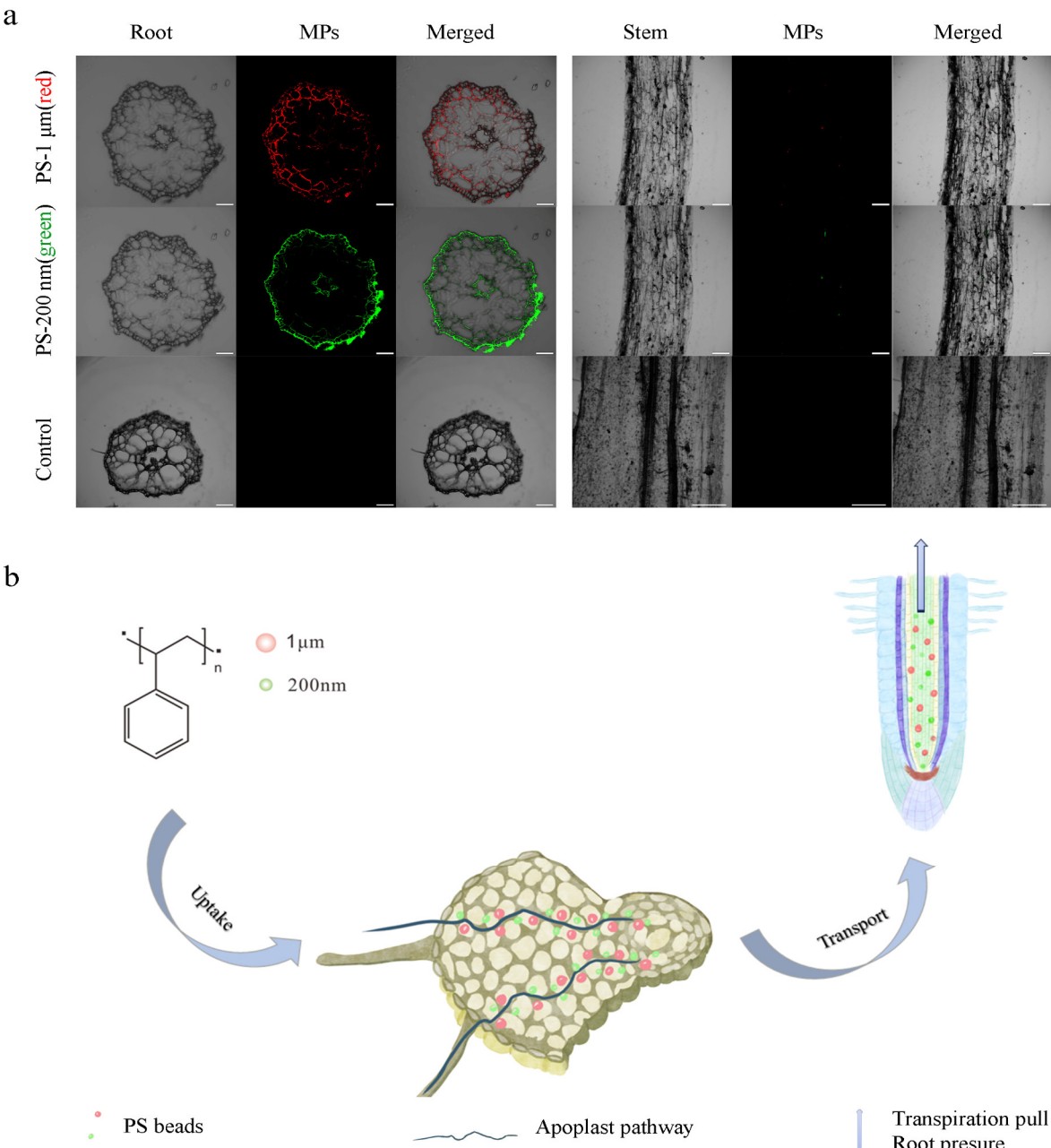

**Figure 1.** Uptake of microplastics and absorption process. (**a**) Laser scanning confocal microscopic images of the root and stem tissues of water spinach grown for 10 days under a polystyrene (PS)—Microplastics (MPs) concentration of 0 mg/L and 50 mg/L. (**b**) The process of MPs absorption by roots. Scale bar: 100 μm.

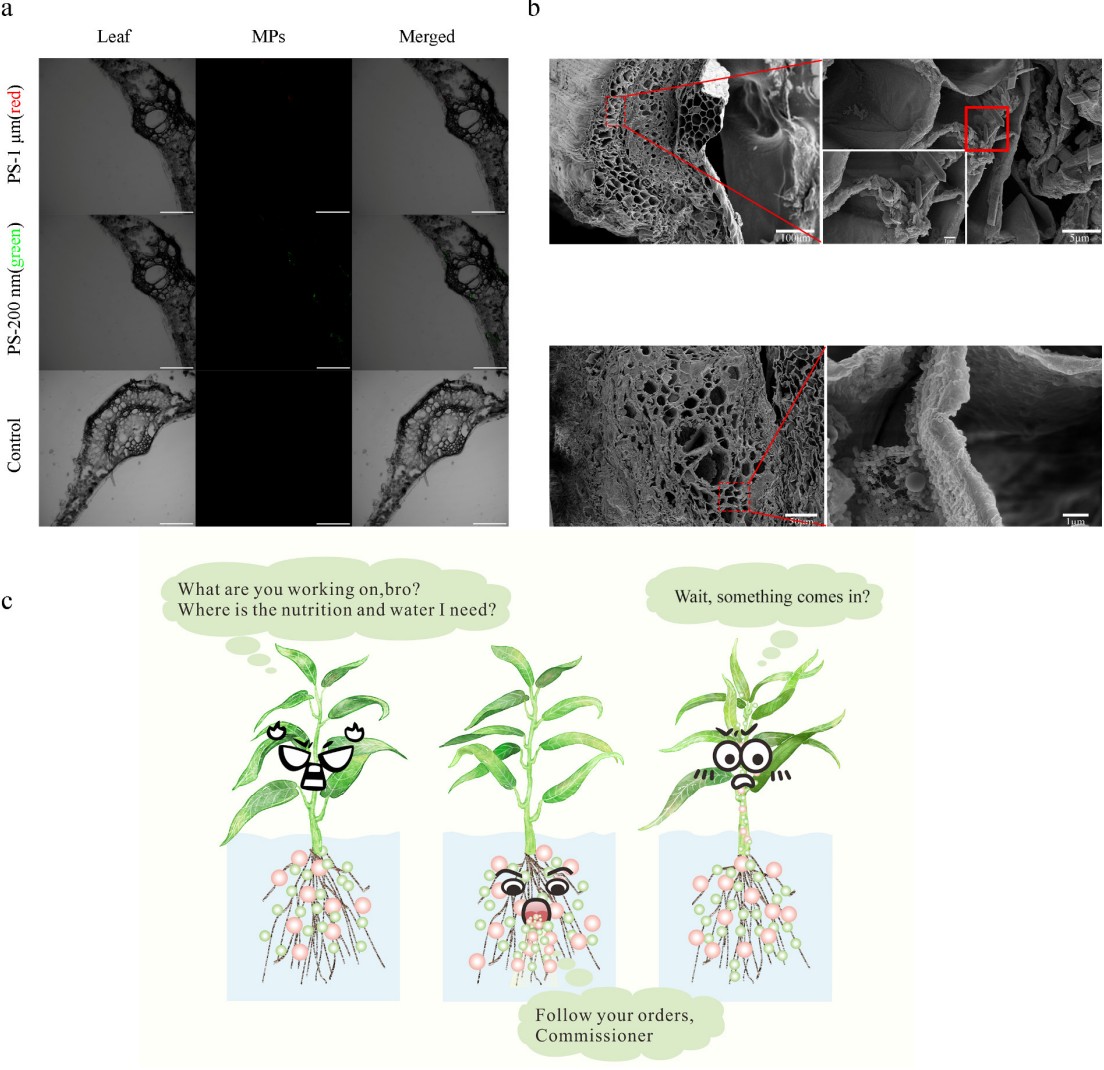

**Figure 2.** (**a**) Laser scanning confocal microscopic images of the leaf tissues of water spinach grown for 10 days following exposure to a mixture of 200 nm and 1 μm PS-MPs. (**b**) Crystal structures in the cells of the leaf vascular tissue; SEM images of the interstitial cell walls of the xylem of the primary root after 10 days of exposure to a mixture of 200 nm and 1 μm PS-MPs. (**c**) Schematic diagram of passive uptake process of PS-MPs in water spinach. Scale bar: 400 μm.

Strong green and red fluorescence signals were observed on the root surface, and vascular cylinder tissues showed equally strong fluorescence signals (Figure S4b,c,e,f). By observing cross-sections of the secondary lateral roots using SEM, 200 nm PS beads were found to be attached to the cell walls of vascular tissues (Figure S5b,d). Thus, this proved that the peripheral root system can indeed absorb submicron-scale PS beads. In addition, we examined the cross-section at the junction between the secondary roots and the primary root using the same method and scanned continuously from the epidermis to the stele. For the first time, we observed both 200 nm PS beads and 1 μm PS beads attached to the cell walls of vascular tissue (Figure S6b,d). In the primary root, no PS beads were found in the control group without the PS bead treatment, while many PS beads were attached to the inner walls of the xylem vessels after treatment with 50 mg L$^{-1}$ PS beads. Both 200 nm and 1 μm PS beads were attached to the cell walls of the vascular tissue in the single-dispersed form (Figure 3d). Moreover, we examined the cortex vascular tissue and found significant differences. In particular, multiple 200 nm PS beads were clustered together with a single 1 μm PS bead (Figure 3f), which was not only observed on the cell walls, but also in the spaces between cell walls.

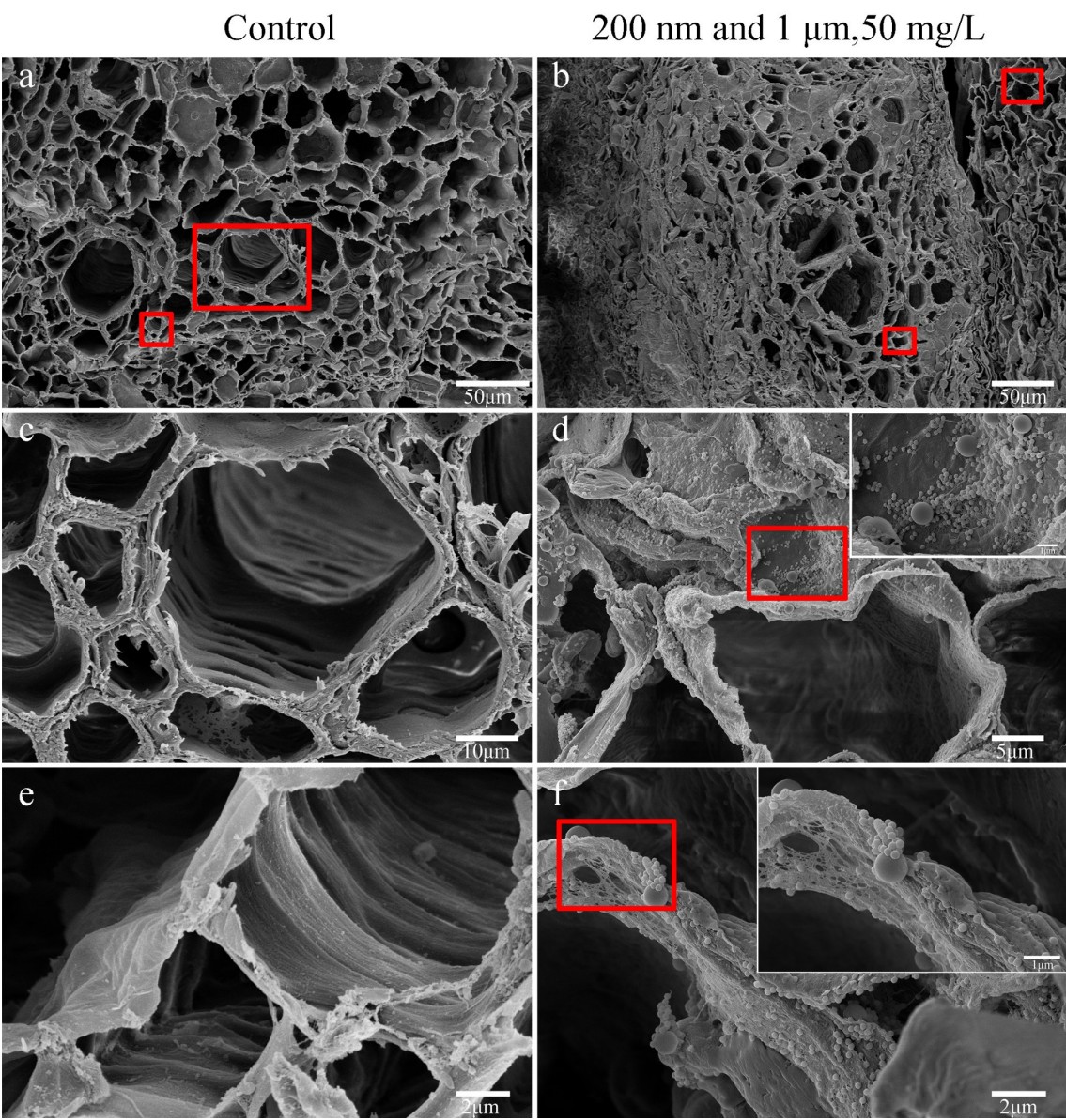

**Figure 3.** Scanning electron microscopic images of transverse sections of fluorescence-labeled vascular tissue of water spinach roots after 10-day treatment with PS-MPs (200 nm, 1 μm, 0 mg/L, and 50 mg/L). (**a,b**) show the overall views of the root stele and the xylem vascular bundles of water spinach. (**c,e**) and (**d,f**) show enlarged views of the red square areas in (**a**) and (**b**), respectively.

Recently, pore structures have been discovered on the surface of corn (*Zea mays* L.) roots, indicating that submicron-sized MPs induce the deformation of corn apical epidermal cells, thus expanding the spaces between the cells and causing deformations and fractures [18]. Similarly, the exposure of Arabidopsis thaliana to PS-NPs changed the shape of root epidermal cells, which led to swellings and deformations in the mature root zone [19]. Entry through such deformations and fractures may be one of the mechanisms responsible for the absorption of PS-MPs. In addition, MPs can enter the root vascular tissue in the lateral root formation region through such microfractures [20]. However, despite the confirmation of many possible routes for MP penetration through the upper epidermis of roots, there are few reports of plants that can absorb micron-scale MPs. Only a few grass species have been confirmed to be able to absorb micron-scale MPs [21], and even large-sized cucumber plants can only absorb submicron-sized MPs [22]. SEM imaging showed that were 200 nm PS beads in the secondary lateral roots of water spinach (Figure S5).

Control          200 nm and 1 μm,50 mg/L

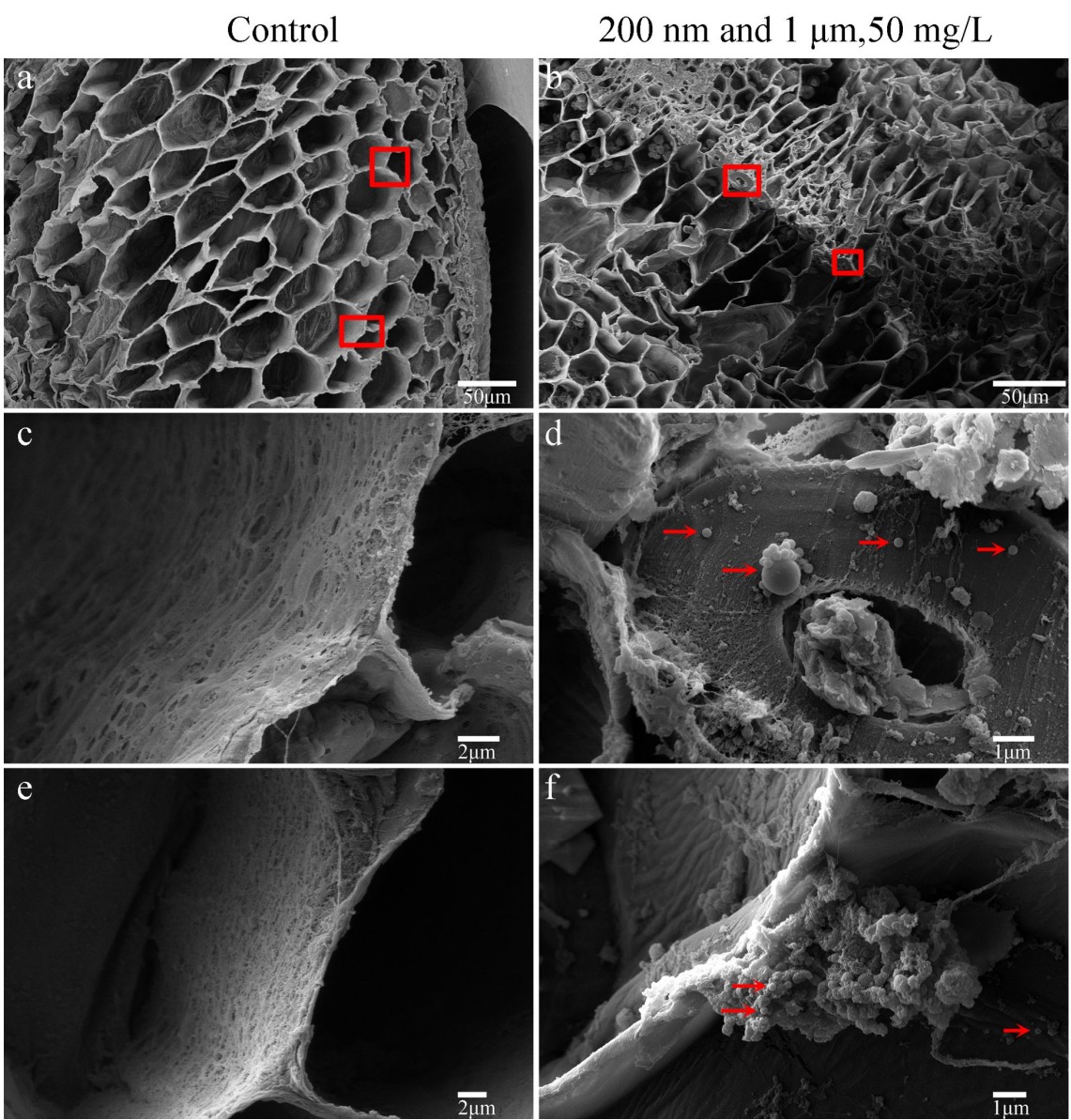

**Figure 4.** Scanning electron microscopic images of transverse sections of vascular tissues of water spinach stems after 10-day treatment with PS-MPs (200 nm, 1 μm, 0 mg/L, and 50 mg/L). (**a**,**b**) show the overall views of the vascular bundles in the stem of an empty leaf, while (**c**,**e**) and (**d**,**f**) show enlarged views of the red rectangular areas in (**a**) and (**b**), respectively. The red arrows in (**d**) point to 200 nm and 1 μm PS-MPs, while the red arrows in (**f**) point to 200 nm PS-MPs.

PS beads of 1 μm were also found in the primary lateral roots (Figure S6), which indicated that the ability of plants to absorb MPs may be related to their root size. The most commonly used experimental approach has been to subject plants to MPs of a single particle size to confirm the upper limit of plant MP uptake. Given the existence of MPs of varying sizes and forms in actual settings [23,24], we believe that this method might not be precise. In this study, we demonstrated that larger-sized MP particles are absorbed in the presence of micrometer-sized MPs. In summary, 200 nm PS-MPs are likely to function as a "key" that causes the deformation of root epidermal cells, which enables micron-sized MPs to enter the root. Thus, this enables the uptake of both sizes of MPs by the root system.

Control              200 nm and 1μm,50 mg/L

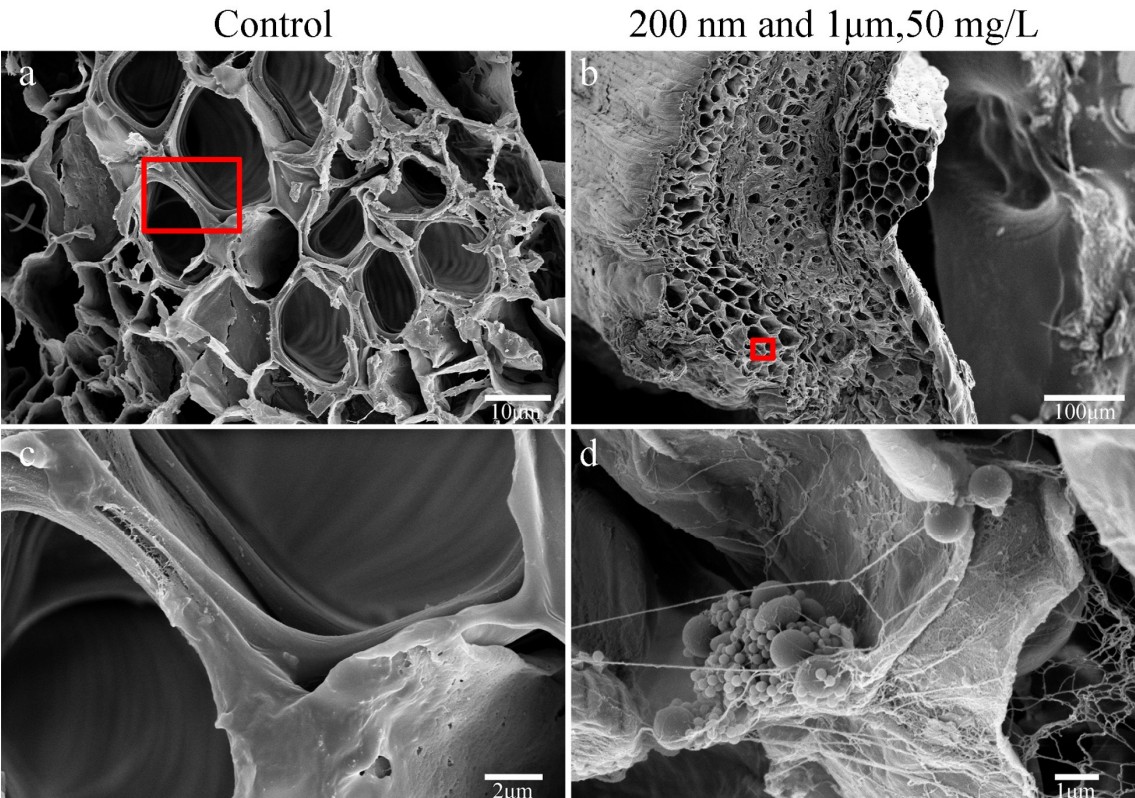

**Figure 5.** Scanning electron microscopic images of transverse sections of vascular tissues of water spinach leaves (including the veins) after a 10-day treatment with PS-MPs (200 nm, 1 μm, 0 mg/L, and 50 mg/L). (**a**,**b**) show the overall appearance of vascular bundles in the cross-section of the leaf blade. (**c**,**d**) show enlarged views of the red boxes in (**a**) and (**b**), respectively.

The cortex is a lateral conduction pathway for transporting water and solutes from the root hairs to the vascular cylinder. In the absence of PS-MPs, fractures are present between the root cortex cells, which can reach up to 1–2 μm, based on TEM images (Figure 6a–c). The MPs can deform and distort the cell walls, which enables the penetration of larger particles and the formation of larger pores [25]. We also observed larger fractures between the root cortex cells, although the possibility that the cell walls wrinkled during the drying process of the sample preparation for SEM cannot be ruled out. However, we observed that 1 μm PS beads were attached to cell walls without deformation (Figure 4b), indicating that once PS beads penetrate the epidermis, they can be transferred to the endodermis through the apoplast pathway.

In the elongation zone of the root tip, the Casparian strip in the endodermis has not yet differentiated (Figure S7d). However, in the maturation zone, the endodermis was completely lignified (Figure S7c). LCSM images showed that both sizes of PS particles were found abundantly inside the elongation zone of the root tip (Figure S3), proving that the PS particles can directly enter the cortex and vascular cylinder by root tip absorption. However, in the mature zone, we found that the PS particles were primarily located in the vascular cylinder (Figure S4). This indicated that the PS particles were primarily transported upwards through xylem vessels by the transpiration flow. Although it is generally assumed that the differentiated endodermis (Casparian strip) strictly regulates the influx and efflux of substances; in fact, endodermis differentiation shows strong plasticity [26]. Nutrient deficiency can lead to the widespread expression of phosphate transporters, thus, reducing the lignification of the endodermis, which provides suitable conditions for PS beads to penetrate the Casparian strip [27]. Short-term hydroponic experiments showed that the PS beads could affect plant growth and the accumulation of biomass (Figure S8). The reason may be that PS beads block cell wall pores, thus, inhibiting water and nutrient

absorption and inducing oxidative damage in plants. These observations are similar to the results of that found that nanoplastics exacerbated the lipid peroxidation damage in water spinach membranes [28]. In addition, we identified crystal structures in the cells of leaves (Figure 2b), which potentially indicate physiological responses of the plant to alleviate stress [29]. Therefore, we suggest that the PS beads can not only directly enter the stele through the root tip, but also penetrate the Casparian strip in the mature zone of the roots.

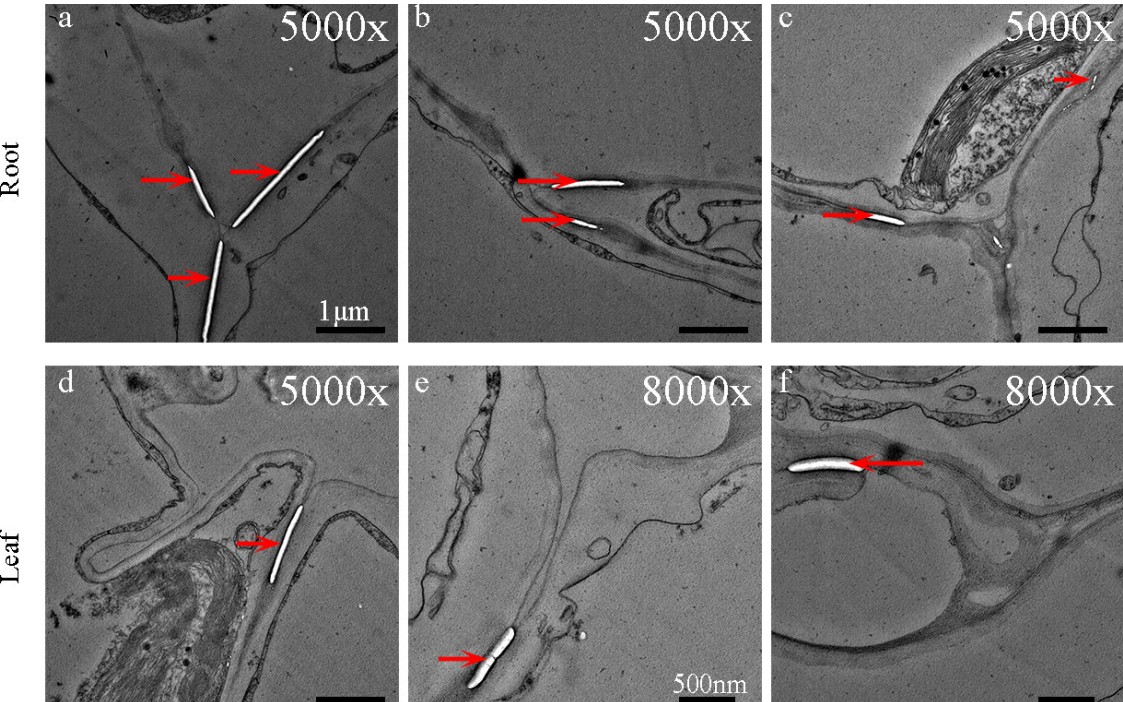

**Figure 6.** Transmission electron microscopic images of cross-sections of the roots and leaves of water spinach grown without added PS-MPs. The numbers in the top right corner of panels (**a**–**f**) represent the magnification factors, and the red arrows point to micro-fractures between cell walls.

### 3.2. PS-MPs Are Transported to the Plant Aboveground Tissues by the Vascular System and Accumulate in Significant Quantities in the Edible Parts of Vegetables

We only observed a small amount of PS bead aggregation in stems (Figures 1a and 4), while a large amount accumulated in the leaves (Figure 5d). This is likely to be related to the unique structure of the water spinach stem. The stem cavity contains a large volume of tissue fluid, and we hypothesize that this tissue fluid is likely responsible for the main transport of water and nutrients. Previous research has shown that the average mercury concentration in water spinach leaves is 1440 $\mu g\ kg^{-1}$, while the average mercury content in stems is only 422 $\mu g\ kg^{-1}$ [30], which further supports our hypothesis. In the leaf vascular system, the cell gaps are smaller than those in the roots (Figure 6), while the cell microfractures are rarely observed in the stems. This also explains why certain micron-sized PS beads were deformed between the leaf cells (Figure 5d). The leaves are the main edible part of leafy vegetables, and the accumulation of a large number of PS-MPs in the leaves poses a substantial risk to human health. Luo et al. [31] demonstrated that wheat seedlings can absorb sub-micrometer plastic particles under soil cultivation conditions, with MPs concentrations as low as 1 mg/kg in the soil. However, the lower accumulation of MPs in stems and leaves also suggests that, compared to aquatic environments, MPs are less likely to come into direct contact with plant roots in soil environments. This may be related to the activity of microorganisms in the rhizosphere [32,33], and the migration ability of microplastics in soil environments is not as strong [34]. However, it cannot be denied that plants still have the ability to absorb and accumulate MPs in soil environments. Moreover, the mutual adhesion of 200 nm and 1 μm PS beads to form larger aggregates more likely

results in the adsorption of higher amounts of heavy metals and other pollutants under natural growth and cultivation conditions in agroecosystems, and should be examined in more detail in future studies.

### 3.3. Leafy Vegetables May Take up Large Amounts of MPs

This study initially proved that leafy vegetables absorb and accumulate micron-sized MPs. It further demonstrated that submicron-sized MPs with a larger specific surface area may induce oxidative stress responses in plants, thereby leading to both the absorption and accumulation of larger amounts of MPs. Notably, green leafy vegetables generally prefer warm and humid environments and have a strong ability to absorb water. This suggests that if MPs are present in the growth environment, it is highly likely that they will be absorbed by the roots. Moreover, because of the continuous increase of the global population, vegetable and lettuce cultivation is increasingly implemented in greenhouse facilities using intensive farming methods [35]. These cultivation practices primarily use plastic products as carriers, such as PVC pipes to deliver fertilizers in hydroponic systems and PS foam boards to support vegetables [36,37]. Additionally, MPs are also present in the organic fraction of compost nutrient products [38]. This presence results in the direct exposure of plants to MPs. Limited by technical means, it is currently impossible to effectively detect nanometer and submicron MPs in the environment, but it is generally accepted that plastics degrade into smaller particles [39]. Based on short-term experiments, this study proved that under the presence of submicron PS beads, micron-sized PS beads are also absorbed in large quantities and accumulate in the edible parts of vegetables.

However, unlike other crops, many leafy vegetables are perennial, such as leek (*Allium ampeloprasum* L.) and water spinach, and they can be cultivated as annual or perennial crops and undergo multiple harvests during this period [40]. As a result, the amount of MPs accumulated will be cumulative across multiple growth seasons after the roots have been initially exposed to MPs during the establishment of the plant. Our current results are based only on short-term observations, and the long-term effects of different types, doses and sizes of MPs on different crops under different growing conditions should be further investigated in practical production applications (soil planting conditions or soilless cultivation modes), in particular, the physiological and molecular effects caused by MPs. This study is of substantial importance for accurately predicting the behavior and trend of microplastics in the "soil-edible vegetable" system and access their environmental and health risks to ensure the quality and safety of edible vegetables.

### 4. Conclusions

To our knowledge, this is the first demonstration of the absorption of micron-scale PS beads by leafy vegetables and provides conclusive evidence that the root system of the aquatic vegetable water spinach can absorb submicron- and micron-scale PS beads and partially transport them to the stem and leaves. When water spinach is simultaneously exposed to different sizes of PS beads, they will be absorbed in large quantities in the short-term. Furthermore, they are transported upward in the stems through an extracellular pathway and eventually accumulate in large quantities in the leaves. The above conclusions are based on hydroponics experiments. When plants are exposed to MPs under soil conditions, the accumulation of MPs in stems and leaves may be reduced, but this does not deny the threat posed by MPs. These results raise concerns about human health. Whether vascular plants exposed to PS beads of different sizes will have a similar situation merits further study.

**Supplementary Materials:** The following supporting information can be downloaded at: https://www.mdpi.com/article/10.3390/agriculture14020301/s1, Figure S1: Properties and characteristics of polystyrene (PS) beads; Figure S2: Water spinach culture in a humus substrate; Figure S3: Laser scanning confocal microscopic images of the root tip; Figure S4: Longitudinally cut laser confocal images of roots of two-week-old water spinach plants grown in a solution with 200 nm and 1 μm PS beads for 10 days; Figure S5: SEM images of transverse sections at the junction of lateral roots

and secondary lateral roots; Figure S6: SEM images of transverse sections at the junction of lateral and primary roots; Figure S7: SEM images of different parts of the root after 10 days of hydroponics cultivation of two-week-old water spinach plants; Figure S8: Growth process of water spinach plants cultured in hydroponics for 10 days.

**Author Contributions:** Conceptualization, methodology, funding acquisition, C.H.; software, validation, formal analysis, investigation, resources, data curation, writing—original draft preparation, writing—review and editing, visualization, project administration, Y.Z.; supervision, funding acquisition, X.W.; supervision, J.X., Y.L., L.W., T.G., A.D., H.C. and Z.W. All authors have read and agreed to the published version of the manuscript.

**Funding:** This work was supported by the National Natural Science Foundation of China [Can Hu, grant number 32060288], the National Natural Science Foundation of China [Xufeng Wang, grant number 32160300], Bingtuan Science and Technology Program [Can Hu, grant number 2021CB010], and the Joint scientific research fund project of Nanjing Agricultural University and Tarim University [Can Hu, grant number NNLH202201].

**Institutional Review Board Statement:** Not applicable.

**Data Availability Statement:** Data will be made available on request.

**Acknowledgments:** The authors would like to thank the Analytical Testing Center at Tarim University for providing the SEM, TEM, LCSM, and other instruments. Eceshi (www.eceshi.com (accessed on 16 May 2023)) is also acknowledged for the SEM experiment.

**Conflicts of Interest:** The authors declare that they have no known competing financial interests or personal relationships that could have appeared to influence the work reported in this paper.

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
