# Peer review of "Water Spinach (Ipomoea aquatica F.) Effectively Absorbs and Accumulates Microplastics at the Micron Level—A Study of the Co-Exposure to Microplastics with Varying Particle Sizes"

_agriculture, doi:10.3390/agriculture14020301_

Round 1
Reviewer 1 Report
Comments and Suggestions for Authors
The authors analyzed absorption and accumulation of submicron-scale and micron-scale microplastics in roots, stems, and leaves of water spinach by exposing it to a mixture of 200 nm and 1 μm polystyrene (PS) beads for 10 days. The results are important in the view of treats which this kind of pollution pose for environment and human health. The authors through series of scanning electron microscopic images proved that water spinach could absorb submicron- and micron-scale PS beads and partially transport them. Very important is also suggestion which comes from results that when water spinach is simultaneously exposed to different sizes of PS beads, they will be absorbed in large quantities in the short-term. The manuscript is quite well written. There are some editorial errors. Suggestions for improvement are below.
General remark:
If possible, could you slightly change the description of the results to make them easier to track? What I mean is that in many places in the Results section you referred to Figures placed in various (sometimes distant) places in the manuscript – for example in sentences in lines 181 – 185: „No fluorescence signals were …. „ you have mentioned: Figure 1a, then Figure 4a, then Figure 2c, e, Figure 3c,e Figure 5c, as well as Figure S2b, c……. Moreover, in the results description you should explain clearly which part refers to the control – 0 mg/L of 200 nm and 1 μm polystyrene and which part is about 50 mg /L dose. This is not always clear from the text unless Reader find Figures, which you refer to. This again make the text difficult to follow.
Other remarks:
Check way of writing latin names of species – they should be in italics – line 50, 55…and in the whole manuscript
There should be also space between a word and brackets with reference numbers – in the whole manuscript
Line 101 – full latin name should be introduced only once – when the organism is mentioned for the first time
Line 276 – „thus” should be after Figure 5 Caption
Line 347 – did you mean really „transportation to the ground”? What do you mean by that?
Reviewer 2 Report
Comments and Suggestions for Authors
In the manuscript by Zhao et al., the investigation of the uptake and accumulation of microplastics by water spinach (Ipomoea aquatica F.) is evaluated.
Indeed, due to the widespread contamination of soil and water by micro and nanoplastics, the absorption of these emerging contaminants by crops is a problem of major concern for human health.
The work presented in this paper investigates the capacity of the water spinach, a leafy vegetable, to effectively absorb and accumulate plastic beads characterised by two different particle sizes: 200 nm and 1 µm, under laboratory conditions.
The authors give a concise but right to the point introduction on the problem at hand.
The experimental workflow is deeply illustrated and detailed; the results are thoroughly discussed and supported by appropriate references.
Overall, this manuscript presents a well-conceived work with results deeply discussed demonstrating the absorption of micron and sub-micron-sized plastic particles by water spinach and the accumulation especially in the edible part of the plant, thus rising health concerns; moreover, the authors propose possible mechanisms of MPs transport within the plant.
Minor comments:
I suggest the authors to briefly expand the introduction addressing the possible adverse effects caused by micro and nano plastics when ingested by humans or animals, hence tackling the problem arising on the possible presence of MPs in edible vegetables.
I suggest the authors to arrange the supplementary figures in the supplementary material according to the order in which they are cited in the manuscript.
Page 2 Line 44: Rephrase the sentence “Once eaten by humans, they can be directly ingested by humans and…” which contains a repetition.
Figure 4: the resolution of figures 4a and 4b should be improved.
Reviewer 3 Report
Comments and Suggestions for Authors
The authors present a study on the mobility of MPs in spinach plants. The work is well structured, but some points must be improved>
1. In the introduction section, the authors should highlight the importance and toxic effects of microplastics. This reference may help.
https://doi.org/10.1016/j.envres.2022.114224
2. The authors used functionalized microparticles to carry out their study, how does their mobility and absorption mechanisms affect microparticles that are not functionalized?
3. The authors must justify in their novelty of the work, how they selected these microparticles? Are they the most frequent in agricultural fields?
4. The authors should improve their conclusion section, highlighting the phenomena or transport mechanisms of microplastics in plant stems, as well as how soil conditions could affect their transport.
Round 2
Reviewer 3 Report
Comments and Suggestions for Authors
The authors responded to the reviewers' comments and their work was improved